# Risk of Intraepithelial Neoplasia Grade 3 or Worse (CIN3+) among Women Examined by a 5-Type HPV mRNA Test during 2003 and 2004, Followed through 2015

**DOI:** 10.3390/cancers15123106

**Published:** 2023-06-08

**Authors:** Amir Rad, Sveinung Wergeland Sørbye, Sweta Tiwari, Maja-Lisa Løchen, Finn Egil Skjeldestad

**Affiliations:** 1Department of Community Medicine, UiT The Arctic University of Norway, 9037 Tromsø, Norway; sweta.tiwari@uit.no (S.T.); maja-lisa.lochen@uit.no (M.-L.L.); finn.e.skjeldestad@uit.no (F.E.S.); 2Department of Clinical Medicine, UiT The Arctic University of Norway, 9037 Tromsø, Norway; 3Department of Pathology, University Hospital of North Norway, 9019 Tromsø, Norway; sveinung.wergeland.sorbye@unn.no

**Keywords:** cervical cancer screening, screening, HPV mRNA test, CIN3+

## Abstract

**Simple Summary:**

Cervical cancer is the fourth most common cancer in women worldwide. Persistent high-risk human papillomavirus (HPV) can cause invasive cervical cancer through a series of precancerous lesions. Screening women with either cytology examinations or a test detecting HPV infection can prevent cervical cancer. HPV tests are substituting cytology examinations in cervical cancer screening, but more knowledge is needed about the screening performance of HPV tests, especially among younger women. We aimed to determine the long-term performance of a five-type HPV mRNA test to predict CIN3+. These results contribute to the knowledge of the reliability of HPV mRNA testing in cervical cancer screening. Our findings suggest that women with a negative result may extend the screening interval up to 10 years.

**Abstract:**

Background: The study’s purpose was to evaluate the performance of a five-type HPV mRNA test to predict cervical intraepithelial neoplasia grade 3 or worse (CIN3+) during up to 12 years of follow-up. Methods: Overall, 19,153 women were recruited by gynecologists and general practitioners in different parts of Norway between 2003 and 2004. The study population comprised 9582 women of these women, aged 25–69 years with normal cytology and a valid five-type HPV mRNA test at baseline. Follow-up for CIN3+ through 2015 was conducted in the Norwegian Cervical Cancer Screening Programme. Results: The cumulative incidence of CIN3+ by baseline status for HPV mRNA-positive and mRNA-negative women were 20.8% and 1.1%, respectively (*p* < 0.001). Age did not affect the long-term ability of the HPV mRNA test to predict CIN3+ during follow-up. Conclusion: The low long-term risk of CIN3+ among HPV mRNA-negative women and the high long-term risk among HPV mRNA-positive women strengthen the evidence that the five-type HPV mRNA test is an appropriate screening test for women of all ages. Our findings suggest that women with a negative result may extend the screening interval up to 10 years.

## 1. Introduction

According to the last global ranking by the World Health Organization (WHO) in 2020, cervical cancer was ranked as the fourth most common cancer in women [1]. High-risk human papillomavirus (HPV) infection is a necessary cause of cervical cancer, which develops slowly over the course of several years through a series of precancerous lesions [2]. By cervical cancer screening of women in the target age group, followed by treatment of detected precancerous lesions and the development of invasive cervical cancer can be prevented. Several randomized trials have shown that cervical cancer screening by HPV testing is more effective than cytology-based screening [3] and several HPV tests have been validated for this purpose. The main differences between HPV tests are which nucleic acid they detect—that of targeted HPV genes (DNA) or that of the transcription of the HPV genome (mRNA)—and the ability of the test to detect and distinguish among HPV types [4].

Cervical cancer is not caused by the HPV infection per se but by the continuous over-expression of the E6 and E7 oncogenes of high-risk HPV types [5]. Expression of the E6 and E7 oncogenes of HPV16, 18, 31, 33 and 45 have been detected in the majority of cervical carcinomas [6]. HPV mRNA tests detect the presence of HPV at the transcriptional level, which indicates continuous expression of the E6 and E7 oncogenes [7]. While both HPV DNA and HPV mRNA tests show high sensitivity to predict cervical intraepithelial neoplasia grade 3 or worse (CIN3+), HPV mRNA tests have a higher specificity than HPV DNA tests [8,9,10]. The lower specificity of HPV DNA tests is more pronounced in younger women [11].

Most longitudinal studies have evaluated the risk of high-grade cervical lesions in women screened with HPV DNA tests. A 10-year cohort study showed cumulative incidence rates for CIN3+ of 17.2% and 13.6%, respectively, in women who were either HPV16+ or HPV18+ at the screening time [12]. However, the corresponding cumulative incidence rate in women positive for all other HPV types was only 3.0%, which was still higher than the rate among women who were HPV negative at the screening time (0.8%) [12]. Another longitudinal study reported 12-year cumulative risks of CIN3+ among women positive for HPV16, 18, 31 and 33 at the screening time of 26.7%, 19.1%, 14.3% and 14.9%, respectively [13].

In Norway, HPV DNA testing started to be implemented for cervical cancer screening in women aged 34–69 years in 2015 but women 25–33 years are still screened with cytology [14]. Since 2023, HPV DNA testing started to be applied every five years for all women aged 25–69 years in Norway [15]. HPV mRNA tests have been reported to be more specific than HPV DNA tests in the triage of women with minor cervical lesions at screening [9,16,17], but there is a gap in the literature and few published studies on the long-term performance of HPV mRNA tests in screening. This study aimed to evaluate the ability of a five-type HPV mRNA test to predict high-grade cervical lesions during approximately 12 years of follow-up. This study is an update of our previous publication on the performance of a five-type HPV mRNA test (PreTect HPV-Proofer, PreTect AS, Klokkarstua, Norway) in screening with histologically confirmed CIN3+ [18] with extended follow-up time. In this update, the “normal” and “unsatisfactory” cytology status were distinguished in our data source.

## 2. Materials and Methods

### 2.1. Data Source and Study Sample

This study received and used data from the Norwegian Cervical Cancer Screening Programme (NCCSP), a division of the Cancer Registry of Norway (CRN), which records the cytology results, HPV results and biopsy results generated in all Norwegian laboratories. Cytology results are classified according to the Bethesda system [19], and histology results are reported applying CIN nomenclature [20]. We applied the 11-digit personal identification number appointed to all Norwegian citizens or immigrants to merge lifetime data on cervical cytology and histology from four national registries administered by the CRN.

Overall, 19,153 women, 13–87 years of age were recruited by gynecologists and general practitioners in different parts of Norway between 1 May 2003 and 31 December 2004. This study was conducted by the departments of pathology and microbiology, University Hospital of Northern Norway, Tromsø. We then excluded women outside the target screening age (younger than 25 and older than 69 years old), women with previous atypical squamous cells of undetermined significance or worse (ASC-US+) cytology, women with a previous histology diagnosis of CIN1+, women with unsatisfactory cytology or no cytology sample collected at baseline and those with no follow-up information. This resulted in a final analytical sample of 9582 women (Table 1). We then followed these women for CIN3+ through 2015.

### 2.2. Screening Guidelines

During the screening period, Norwegian health authorities recommended all women aged 25–69 years be screened with cytology every 3 years [21]. There were no recommendations for HPV testing. The follow-up recommendations during the baseline period were precisely clarified on the website of the CRN [21]. Women with high-grade cytology (atypical squamous cells cannot rule out high-grade lesions/high-grade squamous intraepithelial lesions) were immediately referred to colposcopy and biopsy. Women with ASC-US or low-grade squamous intraepithelial lesions were recommended repeated cytology after 6 months. Women with twice repeated ASC-US/LSIL received triage by repeated cytology after an additional 12 months. Women with 3 times ASC-US/LSIL within 18 months were referred to colposcopy and biopsy. In 2005, delayed HPV triage was implemented in Norwegian screening guidelines. Women with ASC-US/LSIL were referred to repeat cytology and HPV testing after 6–12 months. Women with a positive HPV test and abnormal cytology were recommended for colposcopy and biopsy. Women with a positive HPV test and normal cytology were referred to repeat HPV testing after 12 months. Women with two positive HPV tests in row were referred for colposcopy and biopsy.

### 2.3. Human Papillomavirus mRNA Testing

All HPV mRNA testing conducted during the baseline period took place at the same laboratory. The HPV mRNA test was performed only one time for each woman in the study at the screening time. Among women screened by conventional cytology, an extra specimen was collected and placed in a methanol-containing transport medium (PreTect TM, PreTect AS, Klokkarstua, Norway) for the purpose of HPV mRNA testing. For women screened by liquid-based cytology (LBC), the residual LBC sample preserved in ThinPrep-solution (Hologic Inc., Marlborough, MA, USA) was used for HPV mRNA analysis. From the cervical material, RNA was isolated and preserved in PreTect TM or Thin-Prep medium. HPV mRNA analysis was conducted for all samples by PreTect HPV-Proofer (PreTect AS, Klokkarstua, Norway) according to manufacturer’s instructions. The test detects E6/E7 mRNA transcripts from HPV types 16, 18, 31, 33 and 45 with simultaneous genotype-specific identification including a sample integrity control ensuing sample adequacy. The HPV mRNA testing for women took place between 1 May 2003 and 31 December 2004.

### 2.4. Outcome

The women in the study sample were followed through 2015 for histologically confirmed CIN3+. Only cervical cancer cases that were validated by the CRN against hospital pathology reports were considered true cases of cancer. CIN2, CIN1 and no CIN were considered as absence of disease.

### 2.5. Statistical Analyses

We applied the Pearson Chi-square test for comparisons, Kaplan–Meier survival analyses to show the cumulative status of considered variables during follow-up, and the Wilcoxon (Gehan) statistical method to make pairwise comparisons of categories. 1-survival curves were used to display the cumulative incidence of CIN3+ by HPV status, type and age group.

HPV status was assessed as positive or negative by type: HPV16, HPV18 and HPV31/33/45 (the prevalence of these three types was low, so they were combined). Categorization by HPV type was conducted in hierarchical fashion, i.e., women with multiple infections were assigned to a single category in the following order: HPV16, HPV18 and HPV31/33/45 to reflect the oncogenicity of these HPV types. Age was categorized as 25–33 years and 34–69 years. We had a total of 11 years and 8 months follow-up (mean of 99.8 months), but data are shown for 10 years (120 months), due to the small number of women that remained in the analyses thereafter. All analyses were performed in SPSS version 29.0, and *p*-values < 0.05 were considered significant.

## 3. Results

### 3.1. Human Papillomavirus Status at Baseline

Of the 9582 women included in the present analysis, 3.2% (*n* = 303) were HPV mRNA-positive at baseline; 1.5% (*n* = 140) for HPV16, 0.5% (*n* = 44) for HPV18 and 1.2% (*n* = 119) for HPV31/33/45. Among HPV mRNA-positive women, 46.2% were positive for HPV16, 14.6% for HPV18 and 39.3% for HPV31/33/45. HPV 16 is the most prevalent HPV type, which is followed by HPV45, HPV18, HPV33 and HPV31. At the screening time, 27.2% of women were aged 25–33 years and 72.8% were aged 34–69 years. The HPV positivity rate was 2.6 times higher in the younger than the older age group (5.7% vs. 2.2%; *p* < 0.001; Table 2). Similar age differences were observed for type-specific HPV positivity (*p* < 0.001; Table 2).

### 3.2. Cumulative Incidence of Cervical Intraepithelial Neoplasia Grade 3 or Worse

Of the 303 women who were HPV-positive at the screening time, the cumulative incidence of CIN3+ during follow-up was 20.8% (*n* = 63). Among the 9279 women who were HPV-negative, this incidence was 1.1% (*n* = 104) (*p* < 0.001).

### 3.3. Cervical Intraepithelial Neoplasia Grade 3 or Worse by Human Papillomavirus Type

The cumulative incidence of CIN3+ was alike among HPV16+ and HPV18+ women (22.9% and 22.7%, respectively; *p* = 0.836) (Figure 1). The cumulative incidence of CIN3+ was higher among HPV16+ or HPV18+ women than HPV31/33/45-positive women, but this difference was not statistically significant (15.1%; *p* = 0.118). Among HPV-negative women, the cumulative incidence of CIN3+ was 1.1% (Figure 1).

### 3.4. Cervical Intraepithelial Neoplasia Grade 3 or Worse by Age

The cumulative incidence of CIN3+ among women aged 25–33 years was significantly higher than that among women aged 34–69 years during follow-up (2.2% vs. 1.6%, respectively; *p* = 0.028). After stratifying for HPV status, we observed no statistically significant differences in the cumulative incidence of CIN3+ between the age groups (Figure 2). During the follow-up period, the difference between the cumulative incidence of CIN3+ among women aged 25–33 years and women aged 34–69 years remained insignificant.

### 3.5. Cancer Cases

Five cervical cancers were diagnosed during follow-up; all in women aged older than 38 years. Two were HPV-negative at baseline, one was positive for HPV16, one for HPV18 and one for HPV45. The overall cervical cancer incidence rate was 5 per 100,000 woman years. Cervical cancer incidence rates for HPV-positive and HPV-negative women were 99 (CI: 79.5–118.5) and 2 (CI: 0.8–4.8) per 100,000 woman years, respectively. Three women were diagnosed within the NCCSP, after being referred due to abnormal cytology, whereas two cases were diagnosed during delayed screening or after referral due to the appearance of symptoms. Four cases were in cancer stage I. One case was in cancer stage 2B. Table 3 displays the characteristics of the five cervical cancer cases over the study years.

## 4. Discussion

### 4.1. Cervical Intraepithelial Neoplasia Grade 3 or Worse among Human Papillomavirus-Positive Women

The 10-year cumulative incidence of CIN3+ among the HPV mRNA-positive women was 20.8% (CI: 16.2–25.4). The ability of the five-type HPV mRNA-positive test to predict CIN3+ in our study was higher than the 12-year cumulative risks of CIN3+ among 13-type HPV DNA-positive women in a Danish study (14%) [13] and the 10-year cumulative risks of CIN3+ among 13-type HPV DNA-positive women in a US study (11.3%) [12].

### 4.2. Cervical Intraepithelial Neoplasia Grade 3 or Worse among Human Papillomavirus-Negative Women

The cumulative incidence of CIN3+ within 6 years of follow-up was 0.62% in our previous study [18] while the cumulative incidence of CIN3+ within 10 years of follow-up increased slightly to 1.1% in this study, which can be compared to the 10-year cumulative risk of CIN3+ in 13-type HPV DNA-negative women aged over 16 years [12]. The HPV mRNA test showed equally high sensitivity to predict CIN3+ compared to HPV DNA tests, and it showed a higher specificity than HPV DNA tests [8,9,10]. Considering these findings, it is rational to argue that women with a negative HPV mRNA test result could wait up to 10 years for their next screening.

This is consistent with the 2021 WHO guideline for screening and treatment of cervical pre-cancer lesions, which recommends that the screening interval be extended to 10 years for women with an HPV DNA-negative test [22]. 

Therefore, our data support the effectiveness of the five-type HPV mRNA test as an appropriate screening test for women of all ages. However, further research is required to confirm the long-term effectiveness of the HPV mRNA test and to determine the optimal screening interval.

### 4.3. Age-Independent Ability to Predict Cervical Intraepithelial Neoplasia Grade 3 or Worse

Although the HPV prevalence was higher in the younger than the older age group, the cumulative incidence of CIN3+ was similar among HPV-positive women and HPV-negative women (Figure 2). In a 30 year follow-up study of women screening by HPV DNA test, the highest prevalence of HPV was observed among women aged 25–33 years; however, the risk of cervical cancer was low among women in this age group [23]. Moreover, due to the high HPV DNA positivity rate and low specificity of the HPV DNA test, screening of women younger than 35 years old by HPV DNA test was not recommended [3]. This age-related difference in performance between the HPV mRNA test and an HPV DNA test might be due to what the tests detect on a molecular level. While detection at the DNA level might be infections in a transient phase, with a higher probability of regression, detection at the mRNA level implies an integration phase of the virus into the host genome with more likelihood to progress to a cervical lesion.

### 4.4. Human Papillomavirus Positivity Rate

The specificity of a screening test is an essential characteristic for an effective screening program, as it accurately identifies individuals who will not develop the disease, minimizing false-positive results. A highly specific screening test with a low positivity rate results in a low follow-up rate, which is a critical aspect of screening programs. In this study, we applied a five-type HPV mRNA test in screening, which has been demonstrated to have higher specificity and lower positivity rates than 14-type HPV DNA tests [8,9,24,25]. The overall HPV mRNA positivity rate was 3.2% at screening time, with higher rates observed in women aged 25–33 years (5.7%) compared to those aged 34–69 years (2.2%). This positivity rate was lower than the HPV DNA positivity rates reported in other studies [12,13] such as an American study reporting a positivity rate of 5.1% in women aged 30 years and older [26]. In a European meta-analysis, the overall HPV DNA positivity rate was 9.4% in women aged 20–64 years with variations observed between different countries [27]. The lower positivity rate of the HPV mRNA test in this study is consistent with its higher specificity, suggesting that it may cause fewer false-positive results and a lower referral rate. Although we were unable to determine the sensitivity and specificity of the HPV mRNA test in this study due to the lack of confirmed histological results for all women, its low positivity rate and the observed low long-term risk of CIN3+ among HPV mRNA-negative women support its effectiveness as an appropriate screening test for women of all ages.

### 4.5. Human Papillomavirus Types Included in the mRNA Test

The question of which HPV types should be included in an HPV test used for screening is a matter of discussion. The risk of CIN3+ is strongly associated with persistent infection with HPV16, 18, 31, 33 and 45 [12,13,28,29,30,31,32,33,34]. The HPV mRNA test in our study detects these five high-risk HPV types. A previous report showed that 60.6% of invasive cervical cancers were attributable to HPV16 infection alone, and 70.8% were attributable to HPV16 and/or HPV18 infections [35]. The proportion increases to 84.3% of invasive cervical cancers when expanding the number of high-risk HPV types to five (HPV16, 18, 31, 33 and 45) [35]. A Swedish study examined 808 screen-detected invasive cervical cancers and found six HPV types (16, 18, 31, 33, 45 or 52) in 85.3% of the cases [36]. HPV35, 39, 51, 56, 58, 59, 66 and 68 were detected in only 12 cases (1.5%, for all eight types combined). Therefore, limiting screening to the types included in the five-type HPV mRNA test could greatly improve the specificity of screening programs [36]. The nine-valent HPV vaccine (Gardasil 9) includes HPV6, 11, 16, 18, 31, 33, 45, 52 and 58. Thus, for HPV-vaccinated women, screening tests should be limited to the seven high-risk HPV types (16, 18, 31, 33, 45, 52 and 58), as screening for all 14 HPV types might result in a suboptimal balance of harms and benefits [37]. The results from our study indicate that five HPV types may be sufficient for use in cervical cancer screening.

### 4.6. Cancer Cases

The incidence rates of cervical cancer in HPV mRNA-positive and -negative women were 99 (CI: 79.5–118.5) and 2 (CI: 0.8–4.8) per 100,000 woman per year. The incidence of cervical cancer in women with a negative five-type HPV mRNA test in this study is comparable to that among women with a negative 13-type HPV DNA test in other studies [26,27]. In a meta-analysis of four European countries, the cumulative incidence rate of cancer in women with a negative HPV DNA test at the screening was 2 per 100,000 woman per year [27] at 6.5 years of follow-up. The cumulative cancer rates among HPV DNA-negative women in Italy, the Netherlands, Sweden and England were 0.5, 0.3, 2.9 and 2.6 per 100,000 woman per year, respectively [27]. In an American study including 315,061 women, the incidence rate of cervical cancer was 3.8 per 100,000 woman per year in HPV DNA-negative women [26]. The low incidence rates of cervical cancer after a negative HPV test are in line with the World Health Organization’s strategy and goals to reach and maintain a cervical cancer incidence rate below 4 per 100,000 woman per year [38]. Another point regarding the results of cancer cases in our study is that three to eight years had passed from the HPV mRNA test positive results at the screening time until cervical cancer cases were diagnosed (Table 3). If the women with a positive HPV mRNA test were followed up more closely and received treatment for the precancerous stages before the development of cancer, more cancer cases could be prevented.

### 4.7. Strengths

The NCCSP at the CRN is a nationwide, register-based platform starting in 1995. Compulsory reporting from cytology and pathology departments to the CRN is unique and allowed us to obtain improved information on cytology and histology some years prior to the study’s start, at baseline and during follow-up. Other advantages of this study include the relatively large study sample of women with normal cytology and the long follow-up time.

The HPV laboratory worked independently of cytology laboratories. They were, thus, blinded to cytology results at baseline; cytology laboratories were similarly blinded to HPV mRNA results at baseline.

### 4.8. Limitation

Due to the lack of HPV tests before and/or after the HPV mRNA test at screening, it was impossible for us to know about the persistence of HPV infections before and/or after screening. Another limitation was incomplete screening histories and treatment of CIN before 1995. Determining the sensitivity and specificity of the HPV mRNA test would be desirable; however, the study did not comprise confirmed histological results for all women.

## 5. Conclusions

The low long-term risk of CIN3+ among HPV mRNA-negative women, and the high long-term risk among HPV mRNA-positive women strengthens the evidence that the five-type HPV mRNA test is an appropriate screening test for women of all ages. Our findings suggest that women with a negative HPV mRNA result may extend the screening interval up to 10 years.

## Figures and Tables

**Figure 1 cancers-15-03106-f001:**
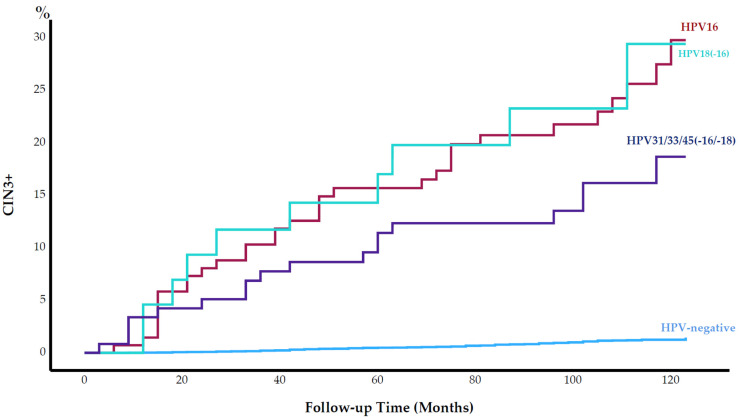
Cumulative incidence of cervical intraepithelial neoplasia grade 3 or worse (CIN3+) (%) by human papillomavirus (HPV) type from baseline throughout 120 months of follow-up.

**Figure 2 cancers-15-03106-f002:**
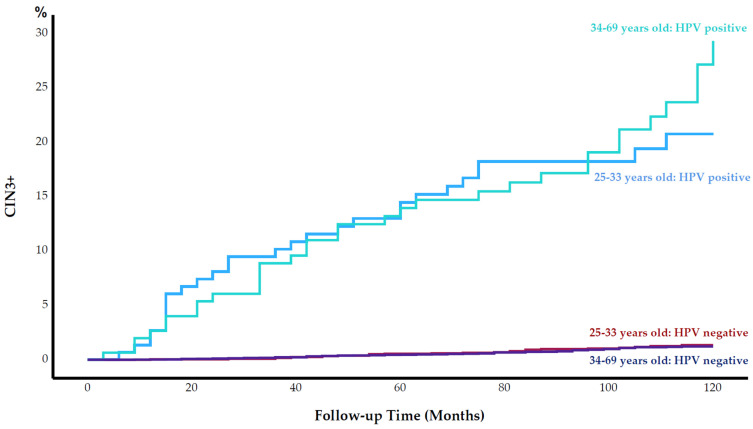
Cumulative incidence of cervical intraepithelial neoplasia grade 3 or worse (CIN3+) (%) by human papillomavirus (HPV) status and by age group during 120 months of follow-up.

**Table 1 cancers-15-03106-t001:** Selection of study sample from 19,153 eligible women. Characteristics at baseline. ASC-US+: atypical squamous cells of undetermined significance or worse, CIN1+: cervical intraepithelial neoplasia grade 1 or worse.

Eligible for Study Participation		19,153
Exclusion Criteria	*n*	
Age < 25 years	2020	
Age > 69 years	223	
Previous diagnosis of CIN1+	883	
Previous ASC-US+ cytology	4756	
Unsatisfactory cytology	501	
No cytology sample collected	627	
No follow-up information	561	
Total exclusions	9571	
**Final study sample**		**9582**

**Table 2 cancers-15-03106-t002:** Human papillomavirus (HPV) status at baseline by age. Positivity rates were significantly different in each row (*p* < 0.001).

HPV Status *	25–33 Years *n* = 2610 (%)	34–69 Years *n* = 6972 (%)	Total *n* = 9582 (%)
HPV Negative	94.3	97.8	96.8
HPV Positive	5.7 *	2.2 *	3.2
HPV16	2.8 *	1.0 *	1.5
HPV18	0.8 *	0.3 *	0.5
HPV31/33/45	2.1 *	0.9 *	1.2

* Categorization by HPV type was conducted in a hierarchical approach, i.e., women with multiple infections were assigned to a single category in the following order: HPV16, HPV18 and HPV31/33/45, to reflect the oncogenicity of these HPV types. Paired comparisons in each row of the table were significant according to Pearson X^2^ test results (*p* < 0.001).

**Table 3 cancers-15-03106-t003:** Characteristics of cervical cancer cases.

Case No.	At Study Start				At Diagnosis			
	Age (yrs.)	HPV Type	Screening History Prior to Study Start	Time to Last Smear Prior to Study Start (Months)	Diagnosed in	Time from Study Start (Months)	Histological Type	Stage
1	38	45	1 normal smear	32	Regular screening	93	SCC	1
2	39	16	8 normal smears	23	Regular screening	38	SCC	1
3	41	18	3 normal smears	34	Delayed screening	58	ADC	2B
4	48	Neg.	4 normal smears	33	Regular screening	28	SCC	1A
5	51	Neg.	6 normal smears	27	Delayed screening	65	SCC	1

SCC stands for squamous cell carcinoma. ADC stands for adenocarcinoma. yrs. stand for years. Neg. stands for negative.

## Data Availability

The data presented in this study are available in this article.

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
