# Peer review of "Risk of Intraepithelial Neoplasia Grade 3 or Worse (CIN3+) among Women Examined by a 5-Type HPV mRNA Test during 2003 and 2004, Followed through 2015"

_cancers, 2023, doi:10.3390/cancers15123106_

Round 1

Reviewer 1 Report

In this manuscript, Amir Rad et al. aimed to determine the long-term performance of a 5-type HPV mRNA test to predict CIN3+. With a large sample, long follow-up, they found that the 5-type HPV mRNA test is an appropriate screening test for women of all ages and women with a negative result may extend the screening interval up to 10 years. The research is properly designed, and the findings showed certain significance. However, there are some flaws and issues that need further clarification. A major revision is suggested.

1.     The method of the study should be presented in detail more clearly. Some key issues need to be specified. Is the HPV mRNA test performed for just one time at baseline period? Did you test for HPVmRNA at 2015? Do all the women go for only regular TCT screening during the 10 years’ follow-up period? For those who were found a cytological abnormality of HSIL or higher, were they excluded from the study for necessary treatment, or they were kept in study to take the histological diagnosis at 2015?

2.     When were the 5 cases of cervical cancer diagnosed? If they were diagnosed during the follow-up period at different time point, how can they be taken to the analysis for “CIN3+ at ten-year period?”

3.     What was the stage of cervical cancer at the time they were diagnosed? What are the status of their cytology and HPVmRNA by the time they were diagnosed?

4.     Did you perform comparative analysis between groups which have different follow-up period? For example, 5 years and 10 years? Probably you will get a lower cumulative incidence of CIN3+ among HPVmRNA negative women. What’s your explanation on this?

5.     Line 62-64: The corresponding cumulative incidence rates in women positive for all other HPV types was 3.0%. If only 5-type HPV mRNA test was performed, how to manage those with persistent infection of other types of HPV?

6.     The results presented in Figure 2 were not well stated in the section of Results. Within 100 months, the cumulative incidence of CIN3+ among women aged 25-33 years was lower than that among women aged 34-69 years during follow-up, however, it changed to be opposite after 100 months, What is the possible reason for this?

7.     Current findings from this study were not strong enough to come to a conclusion that “women with a negative result may extend the screening interval up to 10 years”. At least the sensitivity and specificity should be analyzed.

Author Response

Dear reviewer,

Thank you very much for taking time to review our manuscript.

We really appreciate your comments in improving our paper. We provided our responses to your comments in a separate Word file attached to this letter.

Thank you again for your time and consideration and your valuable suggestions in improving our paper. We hope our updates were fulfilled your concerns.

Best regards,
Rad et. al.

Reviewer 2 Report

This study provides longer term risk estimates for CIN3+ following HPV mRNA testing at baseline, which is important data to have for public health planning and to guide clinical recommendations. I have a couple of comments that may strengthen this article. 

1. Materials and Methods: Is this a convenience sample of women recruited to participate? I'm wondering how representative the women in the sample are compared to the general population. How were medical practices identified in the recruitment phase? Some additional data on this cohort would be helpful.

2. Over the ten year time period, how many women were lost to follow up, died, etc.?

Page 7, line 238: Can you elaborate on why some women didn't have confirmed histological results? I think it should be noted as a study limitation that sensitivity and specificity of the mRNA HPV test could not be determined. 

Author Response

(The authors gave the same response as above.)

Round 2

Reviewer 1 Report

The manuscript has been sufficiently improved.